Uncovering the hidden diversity of Mississippian crinoids (Crinoidea, Echinodermata) from Poland

Salamon Mariusz A. paleo.crinoids@poczta.fm 1
Ausich William I. 2
Brachaniec Tomasz 1
Płachno Bartosz J. 3
Gorzelak Przemysław 4
1 Faculty of Natural Sciences, Institute of Earth Sciences, University of Silesia in Katowice , Sosnowiec , Poland
2 School of Earth Sciences, Ohio State University , Columbus , OH , United States of America
3 Faculty of Biology, Institute of Botany, Jagiellonian University in Kraków , Cracow , Poland
4 Institute of Paleobiology, Polish Academy of Sciences, Poland , Warsaw , Poland
Lieberman Bruce
Electronic publication date: 2020 Dec 21
Publication date: 2020
Volume: 8
Electronic Location ID: e10641
Received 2020 Oct 12; Accepted 2020 Dec 3
Copyright: ©2020 Salamon et al.
Copyright year: 2020
Copyright holder: Salamon et al.
License: This is an open access article distributed under the terms of the Creative Commons Attribution License, which permits unrestricted use, distribution, reproduction and adaptation in any medium and for any purpose provided that it is properly attributed. For attribution, the original author(s), title, publication source (PeerJ) and either DOI or URL of the article must be cited.
License URL: https://creativecommons.org/licenses/by/4.0/

Keywords: Crinoidea, Echinodermata, Taxonomy, Biogeography, Carboniferous

Funding: NCN 2018/31/B/ST10/00387 This project was supported by NCN Grant no. 2018/31/B/ST10/00387. The funders had no role in study design, data collection and analysis, decision to publish, or preparation of the manuscript.

==============================
Partial crinoid crowns and aboral cups are reported from the Mississippian of Poland for the first time. Most specimens are partially disarticulated or isolated plates, which prevent identification to genus and species, but regardless these remains indicate a rich diversity of Mississippian crinoids in Poland during the Mississippian, especially during the late Viséan. Lanecrinus? sp. is described from the late Tournaisian of the Dębnik Anticline region. A high crinoid biodiversity occurred during late Viséan of the Holy Cross Mountains, including the camerate crinoids Gilbertsocrinus? sp., Platycrinitidae Indeterminate; one flexible crinoid; and numerous eucladid crinoids, including Cyathocrinites mammillaris (Phillips), three taxa represented by partial cups left in open nomenclature, and numerous additional taxa known only from isolated radial plates, brachial plates, and columnals. To date, the youngest occurrence of Gilbertsocrinus was the early Viséan of the United States, thus the present finding in upper Viséan extends this genus range. Furthermore, the occurrence of Lanecrinus? sp. expands the Western European range of this genus into the Tournaisian. A single partially disarticulated crown, Crinoidea Indeterminate B, is described from the Serpukhovian of the Upper Silesian Coal Basin. In addition, several echinoid test plates and spines are also reported.

Introduction

Complete aboral cups and crowns of crinoids have not been described previously from Mississippian sediments of Poland. Based on isolated remains Geinitz (1846), Roemer (1870), Schmidt (1930), Korejwo & Teller (1968a) and Korejwo & Teller (1968b), identified Cyathocrinites, Platycrinites, and Poteriocrinites or Ulocrinus. Głuchowski (1980a), Głuchowski (1980b), Głuchowski (1981a), Głuchowski (1981b), Głuchowski (1982), Głuchowski (1986) and Głuchowski (2001) distinguished dozens of columnal taxa from Carboniferous sediments, mainly in southern Poland, using the parataxonomic columnal taxonomy of Moore & Jeffords (1968). Based on columnal taxonomy, Głuchowski (1986) recognized a change in Polish columnal faunas after the Late Devonian. He documented the first crinoid columnal occurrences in Poland from the middle Tourniasian. They were represented by an assemblage with low biodiversity and dominated by isolated remains of Cyclocaudiculus (col.) gracilis Głuchowski. Cyclocaudicus (col.) gracilis remained the main component of the late Tournaisian of western Pomerania (northern Poland). Głuchowski (2001) stressed that the assemblage in the Tournaisian of the Holy Cross Mountains was diverse, being represented by 12 species of which 11 were new. However, as noted all taxa described by Głuchowski (1986) were created based solely on isolated ossicles (no whole or partly preserved cups were recorded). Głuchowski (1986) concluded that this assemblage was endemic and not similar to other late Tournaisian crinoids noted elsewhere in Poland. The most common crinoid taxa in the late Tournaisian of the Upper Silesian Coal Basin were Ampholenium (col.) apolegma Moore & Jeffords, Flucticharax (col.) undatus Moore and Jeffords, and Rhysocamax (col.) cristata Moore and Jeffords. A high crinoid biodiversity was also noted for the Viséan. The most common Viséan columnals were Preptopremnum (col.) rugosum Moore & Jeffords and Cyclocrista (col.) lineolata Moore & Jeffords (Głuchowski, 2001). Crinoid remains were also reported from the Viséan of the Łagów and Kielce Through (southern Poland, Holy Cross Mountains); the Viséan and Namurian of the Upper Silesian Coal Basin; the Viséan, Namurian, and Westphalian of the Lublin Coal Basin (eastern Poland); and from Sudetes (Żakowa, 1956a; Żakowa, 1956b; Żakowa, 1962; Żakowa & Malec, 1992; for a summary see Table 14 in Głuchowski, 2001).

Complete or almost complete Mississippian aboral cups and partial crowns associated with numerous completely disarticulated remains are reported here for the first time from three different exposures in southern Poland (Dębnik Anticline, Holy Cross Mountains, and Upper Silesian Coal Basin).

Stratigraphical setting

Dębnik Anticline

The active Czatkowice Quarry is located near Krzeszowice in the Dębnik Anticline with coordinates 50°09′32.0″N 19°38′17.6″E (Fig. 1D). It is placed along the eastern edge of the post-Hercynian structure, the so-called Sławków Graben. This anticline is filled by Devonian (Givetian-Fammenian) sediments, mainly limestones and dolomites. This sequence is overlain by upper Tournaisian and middle Viséan limestones toward the western part of the anticline (Paszkowski, 2009; Salata, 2013). To the west of the quarry, the Palaeozoic deposits are followed by Triassic and Jurassic sediments. Moving eastward from the quarry, Cambrian to Mississippian strata are covered by Jurassic rocks (e.g., Salata, 2013).

Figure 1 General map of Poland (A) with enlarged maps of the Holy Cross Mountains (B), Upper Silesian Coal Basin (C), and Dębnik Anticline (D) area.

Modified after Marynowski, Salamon & Narkiewicz (2002), Krawczyński (2013), Salata (2013) and Salamon et al. (2018).

The described single crinoid specimen was collected in the early 2000s in a brown limestone layer ∼5cm thick. This limestone was a part of larger carbonate sequence with a thickness of several metres. It belongs to the Mazurowe Doły Formation, which is part of the so-called Rudawa Group (Fig. 2A). The latter formation is a shallowing-upward succession of hummocky cross-stratified and oolitic grainstone that was deposited in a storm-dominated ramp (e.g., Paszkowski et al., 2008). The age of the Mazurowe Doły Formation is late Tournaisian based on co-occurring foraminifera and rugose corals (Poty et al., 2007). Other fossils of this formation are thin-shelled brachiopods, solitary corals, bryozoans, unidentifiable isolated crinoid columnals, gastropods, bivalves, and cephalopods (Thuy, Kutscher & Płachno, 2015 and literature cited therein). The latter authors recorded an articulated ophiuroid specimen of Aganaster jagiellonicus in this formation. According to Paszkowski et al. (2008), the Mazurowe Doły Formation was deposited in a shallow, strongly turbulent subtidal zone with paleo-depths above storm wave base.

Figure 2 Stratigraphic columns of investigated sections. (A) Dębnik Anticline area. (B) Gałęzice area in the Holy Cross Mountains. (C) Gołonóg area in Upper Silesian Coal Basin.

Modified after Belka & Skompski (1988), Paszkowski et al. (2008) and Krawczyński (2013).

Holy Cross Mountains

The active quarry Ostrówka, situated near Gałęzice village, is located in the Kielce Zone of the Holy Cross Mountains with coordinates 50°50′26.5″N 20°24′03.7″E (Fig. 1B). In the Gałęzice area the lithological sequence starts with shallow-water platform carbonates of Frasnian age, which are mainly fine-grained limestones (Fig. 2B). This lithotype is characteristic of Devonian shallow-water environments and is typically interpreted as having been deposited in restricted lagoons between stromatoporoid-coral mounds (Larsen, Chan & Bereskin, 1988). Above the Frasnian deposits is a Famennian pelagic cephalopod limestone. It is a bioclastic wackestone rich in comminuted skeletal debris, containing trilobites, crinoids, ostracodes, and goniatites. The limestone was deposited below the photic zone and storm wave-base. The high content of conodonts suggests a relatively low rate of sedimentation (Bełka, Skompski & Sobon-Podgórska, 1996). Above the Famennian are pelagic carbonates of Tournaisian age. These deposits are mainly limestone breccia and mudstones. Breccia with broken crinoid (pluri)columnals represent the infill of neptunian dykes within the Frasnian host rocks (Bełka, Skompski & Sobon-Podgórska, 1996). Tournaisian mudstones are mostly yellow with rare fossils. This micritic lithology indicates a deep marine, pelagic depositional environment (Bełka, Skompski & Sobon-Podgórska, 1996). Overlying Tournaisian deposits are radiolarian shales with cherts of the lower-middle Viséan Zaręby Beds. Above, the middle-upper Viséan are sediments representing facies equivalent to the Lechówek Beds. This sequence begins with breccias containing clasts of the Frasnian-Viséan rocks, crinoidal limestone, and shales with intercalations of siltstone and sandstone. Most of the crinoids described here (∼99%) are from these late Viséan crinoidal limestones. These sediments are interpreted as gravity flow sediments moved from a shallow platform to a deep basin setting. The age of the Viséan limestones was confirmed by the presence of a diverse foraminiferal fauna dominated by representatives of the genera Endothyra, Howchinia, Valvulinella, Archaediscus, and Tetrataxis (Bełka, Skompski & Sobon-Podgórska, 1996).

Figure 3 Early Carboniferous crinoids from Poland, unless noted otherwise, all specimens are from the Ostrówka Quarry, Holy Cross Mountains. Scale bar equals 10 mm.

A, B. Cladida Indeterminate A, lateral views of both sides of this incomplete crown; note distinctive morphology of the secundibrachials (GIUS 5–3695/Ostrówka 2). C, D. Cladida Indeterminate (B) (GIUS 5–3695/Ostrówka 3), lateral view of aboral cup (C); Oral view of of the radial facets of two radial plates. E. Crinoid Indeterminate A, internal view of the preserved plates from the base of the aboral cup (GUIS 5–3695/Ostrówka 9). F, G. Cyathocrinites mammilaris (Phillips, 1836) (GIUS 5–3695/Ostrówka 1), (F) Lateral view of aboral cup, note large, radial facets; (G) oral view of aboral cup H, Lanecrinus? sp., crown, note stout, long pinnules (GIUS 5–543) (from Czatkowice Quarry, Dębnik Anticline). I. pluricolumnal associated with the crown in (H), this may be a more distal portion of the column of Lanecrinus? sp. (GIUS 5–3695/Czatkowice). J. Archaeocideroid spine boss plate in a coarse crinoidal rudstone. K. Crinoidea Indeterminate B from the Dąbrowa Gónicza, upper Silesian Coal Basin (Serpukhovian) (GIUS 5–3695/Gołonóg 1).

The upper Viséan deposits studied were exposed on the slope in the southeastern part of Ostrówka quarry in 2019. The strata studied was a package of poorly-sorted, coarse-grained crinoidal packstone to rudstone layers, each 30–120 cm thick (Fig. 3J). All these layers contain an extremely abundant and diverse shallow-water benthic fauna: echinoderms, brachiopods, and solitary and colonial corals. These deposits represent material that was transported from an adjacent carbonate platform and deposited in a deeper, lower-slope environment that was part of a submarine, deep-water channelized slope fan (Belka & Skompski, 1988). This has been interpreted as a mixture of faunal elements originating from different ecological niches based on the anatomy of the carbonate lenses, grain-supported texture, chaotic clast arrangement, preferred orientation of elongated bioclasts (rugose corals, crinoid stems), and the presence of reworked fragments from the substrate (Belka & Skompski, 1988; Bełka, Skompski & Sobon-Podgórska, 1996). According to Belka & Skompski (1988), the transport direction appears to be toward the north. So, the source area from which the clast material of the investigated debrite was derived was located south of the Gałęzice area, but the geographical extent cannot be precisely outlined.

Upper Silesian Coal Basin

The historical outcrop in Gołonóg (coordinates 50°19′52.7″N 19°15′33.3″E) is located in northeastern part of the Upper Silesian Coal Basin (Fig. 1C). Here the Pennsylvanian sediments overlie Mississippian mudstones and sandstones of the Culm facies in the western part and Mississippian limestone facies in the eastern part. Mississippian deposits (Serpukhovian; Namurian regional stage) start with the so-called Paralic Series that are represented by mudstones, sandstones and coal seams (Fig. 2C). Above is the so-called Limnic Series (Serpukhovian and Bashkirian) with the Upper Silesian Sandstone Series (Serpukhovian), and the Mudstone Series (Bashkirian). These sediments are represented by mudstones interbedded with sandstones and coal seams. At the top of Carboniferous sediments, mudstones and sandstones of the Kraków Sandstone Series (uppermost Westphalian) are present. These sediments occur only in the eastern part of Upper Silesian Coal Basin (e.g., Krawczyński, 2013).

The described single specimen was found in 2019 within the Gołonóg Sandstones Serpukhovian; early Namurian A age. The Serpukhovian age was determined by Doktorowicz-Hrebnicki (1935), Czarniecki (1959), and Kotas (1972). These sediments belong to Malinowice Beds partly belonging to Marine Diastrophic Series and Paralic Series. Gołonóg Sandstones are located at the boundary of these two series. At present, only a 50 cm thick set of sandstone interbedded with mudstone is exposed in the trail-cut between city districts Tworzeń and Laski of Dąbrowa Górnicza. Within these sediments, external and internal molds of corals, bivalves, gastropods, trilobites, and brachiopods are common (Weigner, 1938; Krawczyński, 2013 and literature cited therein). Salamon (1998) also mentioned disarticulated crinoid columnals and pluricolumnals from Gołonóg.

Material and Methods

The studied material from Czatkowice Quarry (Dębnik Anticline) was collected by Prof. Edward Głuchowski during the early 2000s (two specimens). A single specimen from the Gołonóg sandstones (Upper Silesian Coal Basin) was collected in 2019. The studied material from Gałęzice in the Holy Cross Mountains was collected during the autumn of 2019 and winter of 2020. More than 10,000 columnals and pluricolumnals, a few hundred disarticulated ossicles from aboral cups, arms, columns; and six partially complete crowns/aboral cups were collected in the latter area. The first step consisted of examination of slab surfaces in the field. At this stage, numerous crinoid remains were identified. The next step consisted of soaking seven carbonate samples with Glauber’s salt weighing each from 5 to 7 kg. They were then successively boiled and frozen; depending on hardness of the rock sample, from 1 to as many as 3 times. The residues were finally washed with running tap water and sieved on a sieve column (1.0, 0.315 and 0.1 mm mesh size). The final step consisted of drying the respective washed residues at 160 °C. Residue was hand-picked from each macerated sample for microscopic study. Some specimens were cleaned with hot hydrogen peroxide and then rinsed under running hot tap water.

All larger crinoids were photographed by a Nikon D7100 digital camera, whereas smaller forms by scanning microscope (SEM), a Philips XL–20 at the Institute of Paleobiology of the Polish Academy of Sciences in Warsaw.

The crinoid collection from is housed at the University of Silesia in Katowice, Faculty of Natural Sciences, Institute of Earth Sciences, Poland, under catalogue number: GIUS 5–3695, 5–543.

Results

Systematic paleontology

The classification used herein follows the phylogeny-based revision of crinoid higher taxa by Ausich et al. (2015), Wright (2017a), Wright (2017b), Wright et al. (2017), Cole (2017) and Cole (2018). Morphological terminology follows Ubaghs (1978a), with modifications from Webster (1974), for nodal and internodal terminology), Webster & Maples (2008), Fig. 10.2; for brachial plate terminology), Ausich et al. (2015), and Ausich, Kammer & Mirantsev (2020). All measurements are in mm, with * indicating an incomplete or crushed specimen. Abbreviations for measurements are the following: ACH, aboral cup height; ACW, aboral cup width; AH, arm height; BH, basal plate height; BW, basal plate width; CH, column height; CoH, columnal height; CoW, columnal width; ComaxW, columnal maximum width; CominW, columnal minimum width; CrH, crown height, IH, infrabasal plate height; PbrH, primibrachial height; PbrW, primibrachial width; PCoH, pluricolumnal height, RH, radial plate height; RW, radial plate width; RFW, radial facet width; RmaxW, radial plate maximum width, SBrH, secundibrachial height; SBrW, secundibrachial width.

Class Crinoidea Miller, 1821	
Subclass Camerata Wachsmuth & Springer, 1885	
Infraclass Eucamerata Cole, 2017	
Order Diplobathrida Moore & Laudon, 1943	
Family Rhodocrinitidae Roemer, 1855 (in Bronn & Roemer, 1851–1856)	
Genus GilbertsocrinusPhillips, 1836	
Type species: Gilberstocrinus calcaratusPhillips, 1836	
Gilbertsocrinus? sp.	
Figs. 4A–4C	

Material: GIUS 5–3695/Ostrówka 4–6.

Discussion: Gilberstocrinus (sensu Ubaghs, 1978b) is among the youngest known genera of diplobathrid camerates. It is recognized from Middle Devonian through middle Mississippian (Viséan) strata from Belgium, Canada, China, Ireland, the United Kingdom and the United States. Several morphological aspects of the arms, tegmen, and column of Gilbertsocrinus are unique, which pose interesting questions about their palaeoecology (Van Sant & Lane, 1964; Riddle, Wulff & Ausich, 1988; Hess et al., 1999; Hollis & Ausich, 2008). Columnals from the late Viséan of southern Poland have a morphology that aligns them with Gilbertsocrinus.

Figure 4 Early Carboniferous crinoid columnals from Poland from the Ostrówka Quarry, Holy Cross Mountains. Scale bar equals 1 mm.

(A–C) Gilbertsocrinus? sp. columnal articular facets, note crenulate perilumen and wide areola on all specimens; (A, B) nodals with epifacet (GIUS 5–3695/Ostrówka 5; GIUS 5–3695/Ostrówka 6); (C) internodal without epifacet (GIUS 5–3695/Ostrówka 4). D. pentagonal columnal with crenularium and relatively wide areola (GUIS 5–3695/Ostrówka 50). E. oblique view of a columnal with a narrow crenularium and a smooth latus (GIUS 5–3695/Ostrówka 51). F. oblique view of a columnal with a narrow crenularium and a nodose latus (GIUS 5–3695/Ostrówka 52). G. oblique view of an elongate columnal with a ridge around the columnal at mid height (GIUS 5–3695/Ostrówka 53). H, I. Platycrinitidae Indeterminate columnal, (H) view or articular facet (GIUS 5–3695/Ostrówka 7), (I) oblique view (GIUS 5–3695/Ostrówka 8).

The column construction and columnal facet morphology combine to make Gilbertsocrinus among the most flexible columns known (Lane, 1963; Riddle, Wulff & Ausich, 1988; Hollis & Ausich, 2008). One wide and high columnal alternates with one narrower and lower columnal along the column of Gilbertsocrinus. From the center outward on larger columnals, a columnal facet has a narrow central lumen surrounded by a narrow perilumen with a crenularium. Next is a very wide areola that is surround near the outer periphery by another crenularium. Finally, a narrow epifacet is present around the outer margin of the columnal (Riddle, Wulff & Ausich, 1988). The narrower, shorter columnals have the same morphology, except the epifacet is absent. When the column was in an erect life position, the only contact between adjoining columnals was around the narrow perilumen near the center of the columnals (Riddle, Wulff & Ausich, 1988).

Some upper Viséan columnals from Poland resemble this morphology and are assigned to Gilbertsocrinus? sp. (Figs. 4A–4C). Examples are somewhat worn, which obscures details of the morphology. Also, this occurrence is a range extension for Gilbertsocrinus. Previously, the youngest occurrence of Gilbertsocrinus was the early Viséan of the United States. This occurrence questionably extends this genus range to the upper Viséan.

Measurements: GIUS 5–3695/Ostrówka 4: CoW, 3.1; PCoH, 0.5 (3 columnals). GIUS 5–3695/Ostrówka 5: CoW, 2.8; PCoH, 0.4.

Range: Lechówek Beds (late Viséan); Ostrówka Quarry near Gałęzice, Holy Cross Mountains, Poland.

Order Monobathrida Moore & Laudon, 1943	
Family Platycrinitidae Austin & Austin, 1842	
Platycrinitidae Indeterminate	
Figs. 4H and 4I	

Material: GIUS 5–3695/Ostrówka 7, 8.

Discussion: Ausich & Kammer (2009) recognized that morphological characters that defined platycrinitid genera in Western Europe were not uniformly applied worldwide. Accordingly, Ausich & Kammer (2009) refined genus concepts for the Platycrinitidae, which included a few new genera and many generic reassignments. They assigned many species to Platycrinites sensu lato to refer to species that lack preservation of genus-diagnostic characters.

When first described, the most unique character for Platycrinites (Miller, 1821) was elliptical columns with an articular ridge running across the long diameter of the elliptical columnal. The ridges on the upper and lower facet of a single columnal are offset, which results in the characteristic helical spiral of the platycrinitid column (Wachsmuth & Springer, 1897; Van Sant & Lane, 1964). Historically, middle to late Palaeozoic elliptical columnals have been identified as Platycrinites; however, following generic revisions of Ausich & Kammer (2009), this columnal morphology is recognized as characteristic for the Platycrinitidae in general rather than as Platycrinites.

Only three platycrinitid genera have ranges that include the late Viséan: Eucladocrinus, Platycrinites, and Pleurocrinus. However, only Platycrinites s.l. and Pleurocrinus have described species from the late Viséan of Western Europe, including Platycrinites s.l. conglobatus (Wright, 1937), Platycrinites s.l. crassiconus (Wright, 1937), Platycrinities s.l. invertielensis (Wright, 1942), Platycrinites s.l. murkirkensis (Wright, 1956b), Platycrinites s.l. spiniger (Wright, 1937), Pleurocrinus balladoolensis (Wright, 1938a) and Pleurocrinus vesiculosus (M’Coy, 1849; Ausich & Kammer, 2006; Kammer & Ausich, 2007).

A few platycrinitid columnals are present from the upper Viséan of Poland (Figs. 4H and 4I). Overall, these columnals are relatively small and the height: maximum width ratio (0.6) is relatively high. GIUS 5–3695/Ostrówka 7 has a concave latus, but nodes are present at mid-height around the latus of GIUS 5–3695/Ostrówka 8. It is not possible to confidently assign these specimens to a genus, so they are referred to Platycrinitidae Indeterminate (Figs. 4H and 4I).

Measurements: GIUS 5–3695/Ostrówka 7: CoH, 1.8; CoMaxW, 3.0; CoMinW, 2.0.

Range: Lechówek Beds (late Viséan); Ostrówka Quarry near Gałęzice, Holy Cross Mountains, Poland.

Subclass Pentacrinoidea Wright, 2017a and Wright, 2017b	
Infraclass Inadunata Wachsmuth & Springer, 1885	
Parvclass Cladida Moore & Laudon, 1943	
Cladida Indeterminate A	
Figs. 3A and 3B	

Material: GIUS 5–3695/Ostrówka 2.

Description: Incomplete crown from the radial plates to ∼secundibrachial 12. Crown medium in size, plate sculpturing smooth. Aboral cup shape, infrabasal plates, basal plates, and posterior interray plating unknown. Radial plates ∼1.3 times wider than high; radial facets angustary (∼44% radial plate width). Second primibrachial axillary (Fig. 3A). Secundibrachial width expanded slightly proximally and distally resulting in a broadly concave outline aborally and along the sides of brachials (Fig. 3B). Column unknown.

Discussion: The wide radial facets with angustary radial facets, two primibrachials, 10 total arms, and no apparent ramules or pinnules do not correspond with another Mississippian crinoid. The radial plates are similar to those of Pelecocrinus magnus (Wright, 1937); however, P. magnus is only known from isolated aboral cup plates, and other species of Pelecocrinus are distinct from GIUS 5–3695/Ostrówka 2. Until a specimen is collected with a complete aboral cup, including the CD interray, it is not possible to designate GIUS 5–3695/Ostrówka 2 as either an unusual new species of an existing genus or a new genus. Thus, this taxon is left in open nomenclature at this time.

Measurements: GIUS 5–3695/Ostrówka 2: CrH, 23.0*; RH, 2.0; RW, 2.6; PbrH, 1.0; PbrW, 1.4; SbrH, 1.5; SbrW, 1.6; AH, 20.0*.

Range: Lechówek Beds (late Viséan); Ostrówka Quarry near Gałęzice, Holy Cross Mountains, Poland.

Cladida Indeterminate B	
Figs. 3C and 3D	

Material: GIUS 5–3695/Ostrówka 3.

Description: GIUS 5–3695/Ostrówka 3 is a set of five articulated aboral cup plates: two radial plates, two basal plates and one infrabasal plate (Fig. 3C). The aboral cup shape is either medium or high cone shaped (as preserved). Percentages of plate circlets comprising the aboral cup are ∼13% infrabasal circlet, ∼47% basal circlet, and ∼40% radial circlet. Radial facets are plenary with a straight articular ridge across the entire facet, an aboral ligament fossae and one or two adoral fossae on each side of the adoral groove (Fig. 3D).

GIUS 5–3695/Ostrówka 3 clearly belongs in the articuliformes clade; but without knowledge of the arms and posterior interray plating, this crinoid must remain in open nomenclature.

Measurements: GIUS 5–3695/Ostrówka 3: RH, 4.6; RW, 6.3.

Range: Lechówek Beds (late Viséan); Ostrówka Quarry near Gałęzice, Holy Cross Mountains, Poland.

Superorder Flexibilia Von Zittel, 1895	
Flexibilia Indeterminate	
Figs. 5A–5E, 5G and 5H	

Material: GIUS 5–3695/Ostrówka 12–19.

Remarks: Many flexible crinoids have a patelloid process on their brachials, which is a unique character for the flexible clade (Van Sant & Lane, 1964; Ubaghs, 1978a). The patelloid process is a short, thin extension from the central part of the proximal outer portion of the brachial plate that fits into a corresponding notch on the distal portion of the subjacent brachial plate. In some genera, a patelloid process is also present on the radial plate-first primibrachial articulation. One radial plate with a notch for a patelloid process (GIUS 5–3695/Ostrówka 12, Figs. 5A and 5B) and several brachial plates with a patelloid process are present in washed residues from the late Viséan Lechówek Beds (GIUS 5–3695/Ostrówka 12–19). The articular facets of brachials are unifascial (Figs. 5C–5E) with a narrow crenularium around the margin of the facet and would be termed a synostosis (Fig. 5C). GIUS 5–3695/Ostrówka 14 is an axillary brachial (Figs. 5G and 5H).

Although diminished in biodiversity by the late Viséan, several genera of flexible crinoids were present during this time. Unfortunately, no genus- or species-level traits are present on these radial and brachial plates, and their identification must be left in open nomenclature.

Measurements: GIUS 5–3695/Ostrówka 12 (radial plate): RH, 1.7; RW, 3.4. GIUS 5–3695/Ostrówka 14 (nonaxillary brachial): BrH, 2.6; BrW, 4.7.GIUS 5-3695/Ostrówka 15 (axillary brachial): BrH, 2.8; BrW, 4.5.

Range: Lechówek Beds (late Viséan); Ostrówka Quarry near Gałęzice, Holy Cross Mountains, Poland.

Magnorder Eucladida Wright, 2017a; Wright, 2017b	
Superorder Cyathoformes Wright et al., 2017	
Family Cyathocrinitidae Bassler, 1938	
Genus CyathocrinitesMiller, 1821	
Type species: Cyathocrinites planusMiller, 1821	
Cyathocrinites mammillaris (Phillips, 1836)	
Figs. 3F and 3G	

Material: GIUS 5–3695/Ostrówka 1.

Description: Aboral cup medium globe shape, height to width ratio 0.75; plates convex, smooth. Infrabasal plates presumably five, infrabasal circlet visible in lateral view, ∼22% of the aboral cup height. Basal plates presumably five, hexagonal, convex, smaller that radial plates, ∼32% of the aboral cup height. Radial plates higher than wide, ∼46% of the aboral cup height (Fig. 3F). Radial facets subcircular, strongly declivate, ∼72 percent of distal radial plate width, occupy more than half of the radial facet area; radial facet morphology smooth (Figs. 3F and 3G).

CD interray, arms, and column unknown.

Discussion: The posterior interray on GIUS 5–3695/Ostrówka 1 is not known; but in other aspects, this specimen conforms with the morphology of Cyathocrinites mammillaris. Cyathocrinites mammillaris is a widespread species in Western Europe, having been reported previously from the Tournaisian and lower Viséan of Belgium, Germany, Spain, and the United Kingdom (see Ausich & Kammer, 2006). The isolated radial plates, GIUS 5–3695/Ostrówka 10, 11, may also belong to Cyathocrinites.

Measurements: GIUS 5–3695/Ostrówka 1: ACH, 15.0; ACW, 20.0; IH, 4.0; BH, 6.0; BW, 8.0; RH, 8.5; RW, 8.5.

Range: Lechówek Beds (late Viséan); Ostrówka Quarry near Gałęzice, Holy Cross Mountains, Poland.

Superorder Articuliformes Wright et al., 2017	
Family Decadocrinidae Bather, 1890	
Genus LanecrinusKammer & Ausich, 1993	
Type species: Scaphiocrinus depressusMeek & Worthen, 1870	
Lanecrinus? sp.	
Fig. 3H	

Material: GIUS 5–543.

Description: Crown very small; smooth plate sculpturing, as known. Shape of aboral cup, including posterior interray, not known. Radial facets plenary.

Ten total arms, pinnulate; single isotomous division on first primibrachial. Primibrachials very high, height to width ratio (2.0–3.0). Secundibrachials high, uniserial, moderately cuneate, very high, height to width ratio (2.8–3.0); as many as seven secundibrachials preserved in half-rays. Pinnules articulated to high side of cuneate secundibrachials, long, stout.

Column circular. Proximal portion of proxistele with one wide nodal with a convex latus alternating with one internodal. In distal portion of proxistele, additional cycles of internodals are inserted.

Discussion: GIUS 5–543 is a ten-armed eucladid with a poorly preserved aboral cup; one elongate primibrachial in each ray; elongate, cuneate secundibrachials; prominent pinnules; and a distinctive proximal column. Three aspects of this specimen make it difficult to identify with certainty, including the absence of the CD interray, unknown nature of the aboral cup shape, and the possibility that the small overall size and high brachial plates are indicative of a juvenile specimen.

Known characters of GIUS 5–543 align with Lanecrinus (Kammer & Ausich, 1993). The high brachial plates and robust pinnules (Fig. 3H) are similar to two Western European species of Lanecrinus, L. fifensis (Wright, 1934), Viséan, United Kingdom; and L. trymensis (Wright, 1951b), Viséan, United Kingdom. GIUS 5–543 is recognized herein as Lanecrinus? sp. This occurrence questionably expands the Western European range of this genus into the Tournaisian, but in North America, Lanecrinus is known from the Tournaisian through the Moscovian.

GIUS 5–3695/Czatkowice 2 is an isolated pluricolumnal collected in association with the crown described above (Fig. 3I). This pluricolumnal is preserved in a similar manner to Lanecrinus? sp. (GIUS 5–543), but it is not possible to know with certainty whether these two Tournaisian fossils are from the same species. This pluricolumnal fragment has seven nodals preserved with cirri still attached to four. The column is heteromorphic with a construction of N212. The cirri are long and slender.

Measurements: GIUS 5-543: CrH, 17.2*; ACH, 3.6*; ACW, 3.2*; AH, 14.8*; PbrH, 2.4, 3.6; PrW, 1.2, 1.2; SbrH, 1.8, 2.2; SbrW, 0.6, 0.8; CoH, 7.2*.

Range: Mazurowe Doły Formation, Rudawa Group (late Tournaisian); Czatkowice Quarry near Krzeszowice, Dębnik Anticline, Poland.

Individual crinoid ossicles	
Figs. 4D–4G, 5F and 5I–5V	

Material: GIUS 5–3695/Ostrówka 20–54.

Discussion: Many isolated radial plates are present in the washed residues, and the majority of these are interpreted to be eucladids. In addition to those mentioned previously as potential cyathoformes, five distinctive radial plates are illustrated here. GIUS 5–3695/Ostrówka 20 is a small radial plate with very fine granulose sculpturing and a peneplenary radial facet. Low, broad plications project to subjacent basal plates (Fig. 5F). GIUS 5–3695/Ostrówka 21 has similar very fine granulose plate sculpturing, but it has a flat plate surface and a plenary radial facet (Fig. 5I). GIUS 5–3695/Ostrówka 22 has a protruding, declivate, peneplenary radial facet and fine nodose sculpturing (Fig. 5J). GIUS 5–3695/Ostrówka 24 has a protruding, declivate peneplenary radial facet, but plate sculpturing is smooth (Fig. 5K). GIUS 5–3695/Ostrówka 23 is a radial plate with a small facet with a deep pentafascial, plenary radial facet. This radial plate is distinctive because it has a large spine (relative to the size of the radial plate) projecting abaxially outward (5L, M). Plate sculpturing of GIUS 5–3695/Ostrówka 23 is smooth.

Predictably crinoid plates in the washed residues were dominated by brachial plates, pluricolumnals, and columnals. In addition, a few distinctive brachial plates (GIUS 5–3695/Ostrówka 37–49) and columnals (GIUS 5–3695/Ostrówka 50–54) are noted below. Distinctive brachial plates include GIUS 5–3695/Ostrówka 37, which is a low, weakly cuneate uniserial brachial with a straight articular ridge the full width of the brachial and a very deep adoral groove (Figs. 5O–5P). GIUS 5–3695/Ostrówka 38 is a very high, moderately cuneate uniserial brachial with concave lateral sides, trifascial or pentafascial facets, and articular ridges across the entire facet (Fig. 5Q). GIUS 5–3695/Ostrówka 39 is a low, strongly cuneate, uniserial brachial plate. A small spine projects laterally from the higher side of the cuneate brachial (Fig. 5R). GIUS 5–3695/Ostrówka 40 is a small but very high and narrow rectangular uniserial brachial plate with low serrations along the sides of the brachial (Fig. 5N). GIUS 5–3695/Ostrówka 44 is a large robust axillary first primibrachial plate. All facets have a long, straight articular ridge and are trifascial or pentafascial. This axillary is very similar to the first primibrachials of Hydreinocrinus (e.g., H. goniodactylus) (De Koninck & Wood, 1858) (see Wright, 1951b, pl. 15, fig. 3) (Fig. 5T). GIUS 5–3695/Ostrówka 45 is a uniserial brachial that supported a pinnule. The continuation of the arm projects ∼30 degrees from the axis of the arm, and a distinct, small spine projects laterally below the small pinnule facet (Fig. 5U). GIUS 5–3695/Ostrówka 46 is a high axillary brachial with the width widening at the facets and the sides concave. It is covered by very fine pustulose sculpturing, and a discontinuous ridge is present along the height in the center of the aboral side of the brachial (Fig. 5S). GIUS 5–3695/Ostrówka 47 is a very high, narrow axillary brachial plate with a convex aboral side (Fig. 5V).

Figure 5 Early Carboniferous crinoid crown plates from Poland from the Ostrówka Quarry, Holy Cross Mountains. Scale bar equals 1 mm.

A-C, E, G, H. Flexibilia Indeterminate, (A) outer surface of radial plate, note notch for petaloid process, (B) inside of radial plate illustrated in (A) (GIUS 5–3695/Ostrówka 12), (C) distal facet of brachial plate (GIUS 5–3695/Ostrówka 14), (E) proximal facet of brachial plate with pateloid process projecting out of the image (GIUS 5–3695/ Ostrówka 13), (G) outer view of axillary plate, (H) distal view with two facets on the axillary plate illustrated in (G) (GIUS 5–3695/Ostrówka 15). (D) Radial plate with very fine, granulose plate sculturing and a plenary radial facet (GIUS 5–3695/Ostrówka14). (F) Radial plate with very fine, granulose plate sculturing and a peneplenary radial facet (GIUS 5–3695/Ostrówka 20). (I) Radial plate (GIUS 5–3695/Ostrówka 21). Radial plate with very fine, granulose plate sculturing and a plenary radial facet. (J) Radial plate with a protruding, declivate radial facet (GIUS 5–3695/Ostrówka 22). (K) radial plate (GIUS 5–3695/Ostrówka 24). (L, M) Radial plate with robust spine (GIUS 5–3695/Ostrówka 23). (L) Proximal view with facets to adjacent plates, (distal view illustrating radial facet). (N) Outer view of a very high brachial plate with serrated sculpturing along its margins (GIUS 5–3695/Ostrówka 40). (O, P) Low, weakly cuneate brachial plate, (O) view of facet, (P) outer surface of brachial (GIUS 5–3695/Ostrówka 37). (Q) High, moderately cuneate brachial plates with concave lateral sides (GIUS 5–3695/Ostrówka 38). (R) Spinose, cuneate brachial plate (GIUS 5–3695/Ostrówka 39). (S) Very high axillary brachial plate with strongly concave sides (GIUS 5–3695/Ostrówka 46). (T) Oblique proximal view of an axillary first primibrachial plate (GIUS 5–3695/Ostrówka 44). (U) Brachial plate with a pinnular facet (GIUS 5–3695/Ostrówka 45). (V) Very high axillary brachial plate (GIUS 5–3695/Ostrówka 47).

Despite the large number of columnals and pluricolumnals, few have distinctive features. In addition to the columnals described above as Gilbertsocrinus sp. and Platycrinitidae Indeterminate, four additional columnal morphologies are noted here. GIUS 5–3695/Ostrówka 50 has a pentagonal outline, a crenularium of ∼33% of the facet radius, a wide areola, and a circular lumen (Fig. 4D). GIUS 5–3695/Ostrówka 51 is a columnal ∼2.0 times wider than high with a narrow crenularium, a very wide areola, an elongate lumen that is constricted centrally, and smooth sculpturing on the latus (Fig. 4E). GIUS 5–3695/Ostrówka 52 is a nodal with a similar shape, crenularium, and areola as GIUS 5–3695/Ostrówka 51. It differs by having a circular lumen and distinctive fine nodes on the latus (Fig. 4F). GIUS 5–3695/Ostrówka 53 is a very small, high barrel-shaped columnal with a ridge around the latus at mid-columnal height. Otherwise, the plate sculpturing is smooth (Fig. 4G). This morphology is similar to the flexible crinoid Mespilocrinus (see Wright, 1954a, pl. 67, Fig. 23), but this columnal is much shorter than typical for Mespilocrinus. Further, it is a very small size and may be indicative of a juvenile specimen, and this distinctive shape could be a juvenile morphology rather than the adult morphology of Mespilocrinus.

Measurements: Radial plates: GIUS 5–3695/Ostrówka 20: RH, 2.6; RmaxW, 3.7; RFW, 2.5.GIUS 5–3695/Ostrówka 21: RH, 2.6; RmaxW, 3.7; RFW, 3.5. GIUS 5–3695/Ostrówka 22: RH, 4.5; RmaxW, 5.5; RFW, 3.5. GIUS 5–3695/Ostrówka 23: RH, 2.0; RmaxW, 3.2; RFW, 3.2; GIUS 5–3695/Ostrówka 24: RH, 2.9; RmaxW, 3.9; RFW, 3.1. Nonaxillary brachial plates: GIUS 5–3695/Ostrówka 37: BrH, 2.4; BrW, 6.5; GIUS 5–3695/Ostrówka 38: BrH, 8.3; BrW, 3.9; GIUS 5–3695/Ostrówka 39: BrH, 2.9; BrW, 3.9; BrH, 3.4; BrW, 1.5. Axillary brachial plates: GIUS 5–3695/Ostrówka 44: BrH, 6.6; BrW, 12.5; GIUS 5–3695/Ostrówka 45: BrH, 3.1; BrW, 4.5; GIUS 5–3695/Ostrówka 46: BrH, 4.0; BrW, 1.3; GIUS 5–3695/Ostrówka 47: BrH, 7.7; BrW, 1.3. Columnals: GIUS 5–3695/Ostrówka 50: CH, 0.7; CoW, 3.0; GIUS 5–3695/Ostrówka 51: CH, 1.8; CW, 2.8; GIUS 5–3695/Ostrówka 52: CH, 1.8; CW, 2.9; GIUS 5–3695/Ostrówka 53: CH, 1.4; CW, 1.0.

Range: Lechówek Beds (late Viséan); Ostrówka Quarry near Gałęzice, Holy Cross Mountains, Poland.

Crinoidea Indeterminate A	
Fig. 3E	

Material: GIUS 5–3695/Ostrówka 9.

Discussion: GIUS 5–3695/Ostrówka 9 is a collection of four plates that are an inside view of the basal portion of an aboral cup (Fig. 3E). The four plates are interpreted to be a fused infrabasal circlet and three basal plates. Assuming that the preserved shape of the aboral cup fragment has not been distorted by compaction, this calyx would have had a very gently convex bottom with neither the infrabasal circlet nor most of the basal circlet visible in side view. Viséan dicyclic clades include the diplobathrid camerates, flexibles, and eucladids. This specimen does not appear to conform to the construction of a Viséan diplobathrid in which all adjacent radial plates were separated by intervening plates. The fused infrabasal circlet is a relatively uncommon character for Viséan flexibles and eucladids, and enough morphological information is not preserved to make any further systematic assignment.

Range: Lechówek Beds (late Viséan); Ostrówka Quarry near Gałęzice, Holy Cross Mountains, Poland.

Crinoidea Indeterminate B	
Fig. 3K	

Material: GIUS 5–3695/Gołonóg 1.

Discussion: A single Serpukhovian crinoid is also present. It is a partially disarticulated crown preserved in a sandstone as iron oxide stained casts and some iron oxide molds. No details of the aboral cup can be deciphered beyond it presumably being relatively small compared to the arms. This crinoid presumably had ten robust, pinnulate arms. Brachials are wider than high, moderately cuneate, aborally convex, and diminish in size distally. Pinnulate, uniserial arms indicate that this crinoid is a eucladid, but no distinguishing characters are preserved that allow an identification other than Crinoidea Indeterminate B.

Range: Gołonóg Sandstone (Serpuhkovian); Dąbrowa Górnicza, Upper Silesian Coal Basin, Poland.

Class Echinoidea (Leske, 1778)	
Fig. 3J	

Material: GIUS 5–3695/Ostrówka 56, 57.

Discussion: Several isolated echinoid test plates and spines also occur in the Lechówek Beds. These include test plates from a presumed lepidocentrid and spine boss plates and spines from a presumed archaeocidarid echinoid (Fig. 3J).

Measurements: GIUS Ostrówka–56: archaeocidarid spine boss plate (Fig. 3J), diameter: GIUS Ostrówka–57: lepidocentrid ambulacral plate: plate height, 1.5; plate width, 2.1.

Range: Lechówek Beds (late Viséan); Ostrówka Quarry near Gałęzice, Holy Cross Mountains, Poland.

Discussion

Despite being primarily disarticulated columnal, pluricolumnal, and brachial plates, it is clear that late Tournaisian, late Viséan, and early Serpukhovian crinoid faunas existed in southern Poland. The late Viséan fauna had a high biodiversity and was dominated by cladid crinoids, which is typical for late Viséan crinoid faunas elsewhere (Baumiller, 1994; Ausich, Kammer & Baumiller, 1994; Ausich, Kammer & Mirantsev, 2020), and remains of flexible and camerate crinoids are also present. Isolated brachial plates represent only a minority (c.a. 30%). Selective winnowing may be invoked to explain their scarity. They are primarily from crinoids with uniserial brachial plates and display a high morphological disparity that reflects a high biodiversity. Consistent with the brachials, a high morphological disparity is present in isolated radial plates, columnals, and pluricolumnals. However, without a fauna with complete specimens preserved, it is not possible to identify most of these individual plates beyond higher taxonomic clades.

As described in more detail above, one aboral cup, two fragmentary aboral cups, and three partial crowns (all cladids) were recovered. Knowledge of several key morphological characters are necessary to identify most crinoid genera and species. For cladids, one typically must know the aboral cup shape, the height of the radial plate circlet compared to the aboral cup height, radial facet types, the number and arrangement of posterior interray plates, the number of primibrachials, shape of the brachials, and arm branching pattern.

Rather than the column-based taxonomy of previous studies (Głuchowski, 1980a; Głuchowski, 1980b; Głuchowski, 1981a; Głuchowski, 1981b; Głuchowski, 1982; Głuchowski, 1986; Głuchowski, 2001), this study uses crown-based taxonomic names. Among Carboniferous crinoid faunas, crown-based taxa are typically very diverse, whereas column disparity is typically low. Further, columnal morphology can change along a single column. There has been little work to reconcile current column-based taxonomy with crown-based taxonomy on Carboniferous crinoids. However, two very distinctive columnals are assigned to crown-based taxa, including Gilbertsocrinus? sp. and Platycrinitidae Indeterminate. Echinoids were also present in the Lechówek Beds. Because morphological disparity of Mississippian crinoid columns is quite low, this disparity is a poor reflection of the overall crinoid biodiversity and it is not possible to recognize many crinoid columnals with column-based taxonomic names. Although brachial plate disparity in the fauna is also not a true indication of biodiversity, it should be a much more accurate reflection of biodiversity than isolated columnals. Crinoid arm morphology is a key, commonly species-specific attribute that changes the crinoid filtration fan density, thus defining niches among crinoids (Ausich, 1980; Cole, Wright & Ausich, 2019).

The descriptions above reveal that diverse Mississippian faunas existed in present-day southern Poland. Most of the crinoidal remains are disarticulated and cannot be identified, but this study demonstrates that continued fieldwork holds promise for discovery of many new specimens that will yield a better understanding of crinoid faunas from the late Tournaisian of the Dębnik anticline region, the late Viséan of the Holy Cross Mountains, and the Serpukhovian of the Upper Silesian Coal Basin. The preservation of partial crowns in all of these settings indicates that depositional conditions were present for excellent crinoid preservation, and the discovery of additional specimens should be expected. Lanecrinus? sp. is described from the late Tournaisian the Dębnik anticline region, and Crinoidea Indeterminate B is described from the Serpukhovian of the Upper Silesian Coal Basin. Remains of a substantial fauna is described from the late Viséan of the Holy Cross Mountains, including Gilbertsocrinus? sp., Platycrinitidae Indeterminate, Cyathocrinites mammillaris (Phillips, 1836), a flexible crinoid, and partial aboral cups of three eucladids. In addition, radial plates, brachial plates, and columnals described below indicate a much more diverse fauna, as exemplified by the description of five distinct radial plates; eight distinctive brachials; and in addition to Gilbertsocrinus? sp., Platycrinitidae Indeterminate, four distinctive columnal morphologies are described.

Supplemental Information

Supplemental Information 1 Specimen numbers

The Carboniferous crinoid collection from Poland is housed at the University of Silesia in Katowice, Faculty of Natural Sciences, Institute of Earth Sciences, Poland, under catalogue number: GIUS 5–3695, 5–543.

Click here for additional data file.

We thank Ms. Karolina Paszcza, Mr. Tomasz Borszcz, Mr. Bartosz Czader, and Mr. Mateusz Syncerz, for help during the field works and donating some of specimens to present investigations. We also thank Jeffrey R. Thompson for his advice on identification of the echinoids; and James R. Thomka and an anonymous reviewer, who provided careful reviews that helped us improve this paper.

Additional Information and Declarations

Competing Interests

Author Contributions

Data Availability

The authors declare there are no competing interests.

Mariusz A. Salamon and William I. Ausich conceived and designed the experiments, performed the experiments, analyzed the data, prepared figures and/or tables, authored or reviewed drafts of the paper, and approved the final draft.

Tomasz Brachaniec analyzed the data, authored or reviewed drafts of the paper, field works, and approved the final draft.

Bartosz J. Płachno performed the experiments, analyzed the data, authored or reviewed drafts of the paper, field works, and approved the final draft.

Przemysław Gorzelak performed the experiments, analyzed the data, prepared figures and/or tables, authored or reviewed drafts of the paper, and approved the final draft.

The following information was supplied regarding data availability:

The Carboniferous crinoid collection from Poland is housed at the University of Silesia in Katowice, Faculty of Natural Sciences, Institute of Earth Sciences, Poland, under catalogue number: GIUS 5–3695, 5–543.

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
