# Peer review of "Uncovering the hidden diversity of Mississippian crinoids (Crinoidea, Echinodermata) from Poland"

_PeerJ, doi:10.7717/peerj.10641_

## Round 0.1 · original submission · Minor Revisions

The reviewers have provided useful comments, and these should be pretty straightforward to make. When you resubmit, please provide a letter stating how you have addressed the reviewer comments. Also, include a version of your revised manuscript with the changes tracked.

Reviewer 1 ·

Basic reporting

This is a thorough and detailed description of newly discovered crinoid taxa from the Carboniferous of Poland. It is a much needed look at these taxa, since previous crinoid diversity in the region has only been known from columnal-based taxa, which, as the authors discuss, under represents true diversity in the region. Since finding adequately articulated material for crown-based taxonomy is no easy task, I believe these findings are of significant importance for understanding Carboniferous crinoid diversity. The systematic treatment of the material is thorough, and I find the imaged to be of high quality.

Experimental design

The systematic treatment of these crinoid specimens is rigorous, and the descriptions and imaged provided are highly detailed.

Validity of the findings

I am not an expert of crinoid systematics and therefor cannot comment on the specifics of their assignments.

Additional comments

This is only a minor suggested for a figure enhancement:

Though the description of the stratigraphic setting is highly detailed and thorough, I would have also liked a visual aid of a generalized/composite stratigraphic columns representing the 3 localities studied, with indications of where the crinoid specimens came from. Perhaps something like this might work well as a part of Figure 1?

·

Basic reporting

This is a well-written and well-referenced contribution, with proper attention paid to previous studies of regional crinoid faunas. The structure of the manuscript is logical, in keeping with established conventions for systematic descriptions. I commend the protocols for keeping certain taxa in open nomenclature rather than omitting their descriptions entirely or assigning them to an overly speculative taxonomic identity. All relevant raw data, including museum specimen numbers and measurements, are clearly provided, as are justifications for identifications (or lack thereof) of incomplete material.

Areas where some revision is warranted include the following:

1. Figure 1 and associated captions are unclear. It should be made clear that panel A is an overview map of Poland with the regions depicted in the other panels denoted by corresponding letters. Or, even more effectively, the outlines of the geologic regions shown in panels B-D should be included in the map of Poland in panel A. Also, in keeping with journal formatting, the dates for the references in the captions should be in parentheses.

2. In the "Stratigraphical setting" section, the number of specimens collected from each site should be mentioned. This will allow readers to more fully link the geologic setting to the detailed specimen descriptions provided later in the manuscript. Nearly ALL of the material described in this manuscript (10,000+ specimens) come from the Holy Cross Mountains, with only a single specimen described from the Debnik Anticline and only a single specimen described from the Upper Silesian Coal Basin. This distribution of material should be made clear in the beginning of the paper.

3. On line 86, it is stated that crinoids had been previously reported from the Mazurowe Doly Formation. The nature of that crinoid material should be made clear (i.e., isolated crinoid columnals vs. articulated material; taxonomically identifiable vs. unidentifiable material).

4. Should the descriptions for Cladida indeterminate A and Cladida indeterminate B moved to an earlier position in the systematic section? It seems logical that they would immediately follow Parvclass Cladida on line 273.

Experimental design

The methods employed are very clear and thoroughly explained, and the findings presented in this study are appropriate for publication in this journal. The research question is stated clearly, and the novel data fill a gap in the existing knowledge of Polish crinoid faunas. Methods are described in sufficient detail to permit replication.

Areas where some revision is warranted include the following:

1. On line 167, please use only mm for describing the size of mesh. If the symbol before "1.0" is supposed to be phi, then this is an incorrect symbol.

2. On lines 170-171, it is stated that 19 polished slabs were prepared for examination, but are not described, figured, or discussed further in this report. What is the significance of these polished sections---do they contain any relevant data?

Validity of the findings

The data presented in this study are well-supported and fit with the most up-to-date knowledge of crinoid morphology and taxonomy. Conclusions are linked to detailed descriptions and figured specimens.

Areas where some revision is warranted include the following:

1. The high morphological disparity of brachial plates is used as evidence of high biodiversity on lines 526-527; however, the fact that columnal morphology varies along a column is stated on line 540. Brachial morphology also varies along crinoid arms. Are you stating that morphological diversity is a reflection of biological diversity for brachials but not for columnals?

2. This is not a criticism, nor is it a required change, but I am curious about the abundance of columnal/pluricolumnal material relative to brachial material. I've worked with several deposits--both carbonate and siliciclastic--that are dominated by isolated ossicles, but with far greater numbers of brachials compared to columnals. Other crinoid researchers have also asserted that brachials are typically the dominant crinoid bioclast, but that this pattern is poorly documented. Anything that you can add to this manuscript addressing this pattern, whether it be an issue of perception or a genuine bias in the crinoid fossil record, would be very welcome to those of us who work extensively with disarticulated crinoid material

Additional comments

I commend the authors on this detailed, well-written, and well-illustrated study. Research focusing on "ugly" crinoidal material, including isolated ossicles and fragmentary specimens, is quite important given the echinoderm fossil record and can provide significant insights into depositional processes, paleobiogeographic distributions, paleoecology, and taphonomy. However, this material is far too often ignored or given hasty treatment. Studies such as this should be strongly encouraged, as easily overlooked specimens can generate valuable information on echinoderm diversity (taxonomic and/or morphologic) and occurrence.

Please feel free to contact me if you have questions or if there is anything that I can provide greater clarity on. Thank you,

James R. Thomka*

Center for Earth and Environmental Science
State University of New York at Plattsburgh
Plattsburgh, New York 12901, USA

*jthom059@plattsburgh.edu

---

## Round 0.2 · accepted · Accept

The authors did a very good job of addressing the comments from the reviewers and the paper is ready to go to the next stage of production.